# A New Direction for the Green, Environmentally Friendly and Sustainable Bioproduction of Aminobenzoic Acid and Its Derivatives

Shujian Xiao [1], Rumei Zeng [1], Bangxu Wang [1], Suyi Zhang [2,3], Jie Cheng [1,2,3,*] and Jiamin Zhang [1,*]

1  Meat Processing Key Laboratory of Sichuan Province, College of Food and Biological Engineering, Chengdu University, Chengdu 610106, China; xiaoshujian1995@163.com (S.X.); zengrumei@stu.cdu.edu.cn (R.Z.); wangbangxu@stu.cdu.edu.cn (B.W.)
2  Postdoctoral Research Station of Luzhou Laojiao Company, Luzhou Laojiao Co., Ltd., Luzhou 646000, China; zhangsy@lzlj.com
3  Solid-State Brewing Technology Innovation Center of Sichuan, Luzhou 646000, China
*  Correspondence: chengjie@cdu.edu.cn (J.C.); zhangjiamin@cdu.edu.cn (J.Z.)

**Abstract:** Aminobenzoic acid and its derivatives are a class of aromatic compounds that are important foundational chemicals for various dyes, food additives, and pharmaceuticals. Their production relies on chemical synthesis using petroleum-derived substances such as benzene as precursors, but due to the toxicity, environmental pollution, and non-renewable nature of raw materials in chemical synthesis, some suitable alternative methods are gradually being developed. Green, environmentally friendly, and sustainable biosynthesis methods have gradually been favored by researchers, especially after the discovery of the synthetic pathways of aminobenzoic acid and its derivatives in plants and microorganisms. Based on the purpose of protecting the ecological environment, reducing the use of non-renewable resources, and providing theoretical support for industrial green development, this article reviews the biosynthesis pathways of *ortho*-aminobenzoic acid, *meta*-aminobenzoic acid, *para*-aminobenzoic acid, and its derivatives such as catechol, folic acid, etc., and lists some examples of biosynthesis, analyzes their advantages and disadvantages, summarizes and looks forward to the future development direction of biosynthesis of aminobenzoic acid and its derivatives.

**Keywords:** aminobenzoic acid; biosynthesis; shikimate pathway; environmental pollution; green industry

## 1. Introduction

Aminobenzoic acid (ABA) is an aromatic amino acid containing a benzene ring, with a molecular formula of $C_7H_7NO_2$. Based on the position of the amino group, ABA has three structural formulas, namely *ortho*-aminobenzoic acid (OABA)/anthranilate (ANT), *meta*-aminobenzoic acid (MABA), and *para*-aminobenzoic acid (PABA), which are isomers of each other. They are also known as 2-aminobenzoic acid, 3-aminobenzoic acid, and 4-aminobenzoic acid (2-, 3-, and 4-ABA) [1]. They are non-proteinogenic, but their bifunctional nature allows them to be linked to other scaffolds and utilized in various niches of microbial metabolism [2]. OABA is widely present in the biochemical pathways of animals, plants, and microorganisms. It serves as a precursor for certain compounds in nature, including alkaloids and indole-3-acetic acid [3]. PABA is a key intermediate in the biosynthesis of folate [4]. MABA is an essential pharmaceutical intermediate and it also finds significant applications in the synthesis of novel organic luminescent materials and plant growth regulators [5]. In addition, derivatives of ABA have been found to possess antibacterial and insecticidal properties. For example, sulfanilamide, a derivative of PABA, is widely used for the prevention and treatment of infectious diseases [6]. In short, ABA and its derivatives are an important class of organic compounds. They possess diverse

biological activities and have wide-ranging applications in the fields of pharmaceuticals, agriculture, and chemical industry.

ABA and its derivatives have extensive applications in various industries. The global market size of OABA alone is expected to reach EUR 120 million by 2024 [7]. With such a large market size, the demand for their production will also be significant. In the past, these substances were mostly synthesized from petroleum derivatives (such as phthalamic acid) through energy-intensive chemical methods [8]. The petrochemical industry is widely known to involve a large number of highly flammable, corrosive, toxic, and explosive substances. Additionally, it can cause environmental pollution and energy waste and contribute to climate change, among other adverse effects [9]. According to statistics, the chemical industry is one of the highest energy-consuming and most severe polluting industries in China [10]. The industrial wastewater and exhaust gases from the petrochemical industry contain a large amount of toxic and harmful substances such as benzene compounds and heavy metals (Figure 1). These pollutants can spread to the surrounding environment through surface water, groundwater, and air, leading to contamination of land, water sources, and air, thereby severely impacting the surrounding ecological environment [11–13]. Pollution can render land unsuitable for plant growth, result in the death of fish in water bodies, and pose serious health risks to humans living in the vicinity. Prolonged exposure to harmful environments and contact with pollutants can significantly jeopardize physical well-being, leading to an increased risk of cancer, aplastic anemia, and neurological disorders [11,14]. The damage caused by such pollution to the environment and ecosystems is generally considered irreversible. Furthermore, fossil resources are non-renewable and excessive reliance on and exploitation of these resources will eventually lead to fuel exhaustion. Therefore, green, ecological, and sustainable synthesis is the trend for the future [15–17].

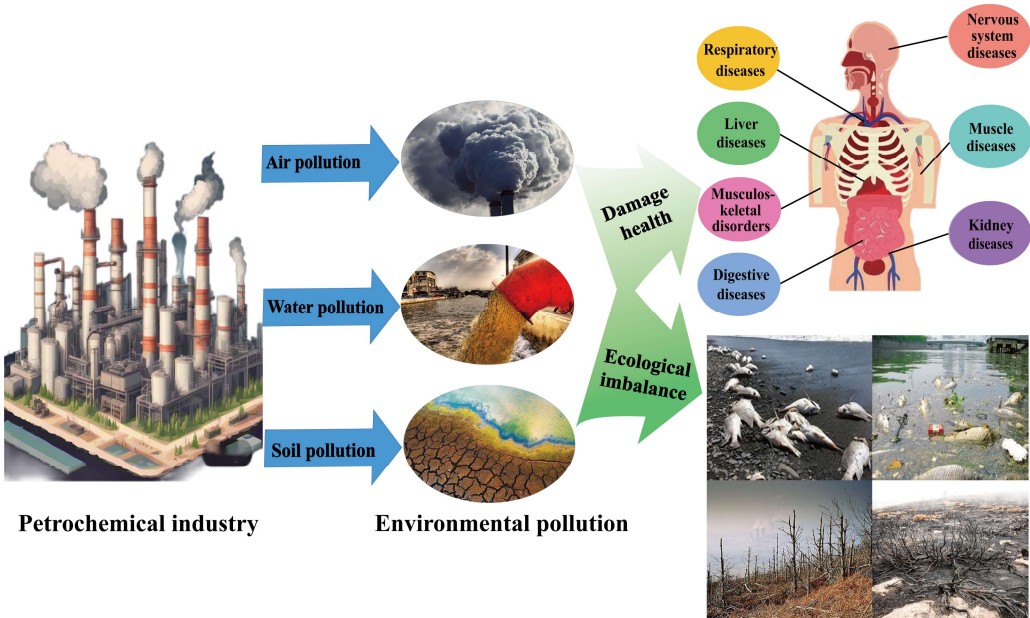

**Figure 1.** Hazards of petrochemical industry.

The metabolic pathways of ABA and its derivatives are present in various plants and microorganisms and are important components of the shikimate pathway [18]. They mostly use glucose, xylose and other carbon sources to produce ABA and its derivatives in their own life activities. This means that the use of microorganisms as hosts to produce ABA and its derivatives can theoretically replace chemical synthesis to a certain extent, thereby reducing the emission of industrial waste and the use of fossil raw materials [19]. The traditional chemical synthesis of ABA and its derivatives requires multi-step synthesis using petroleum extracts as precursors, using catalysts, under acidic or alkaline, high

temperature and other conditions. For example, the synthesis of OABA requires the use of $V_2O_5$ to catalyze the oxidation of o-xylene to obtain phthalic anhydride, which is opened by ammonia to form phthalamide, and then treated with sodium hypochlorite to obtain OABA. These steps are not required in biosynthesis. Therefore, utilizing renewable resources to produce ABA and its derivatives based on microorganisms is a good and environmentally friendly alternative method. In fact, biocatalytic methods can even solve the problem of large amounts of waste generated by some biorefineries, achieving waste utilization and protecting the ecological environment [20]. This article summarizes the research progress on the synthesis of ABA and its derivatives in various microorganisms, summarizes their metabolic pathways, and provides prospects for future development. The aim is to provide theoretical support for using biological production to replace the petroleum-based production of ABA and its derivatives, and to make a contribution to environmental protection.

## 2. Synthesis Pathway

### 2.1. OABA and Its Derivatives

The synthesis of OABA, MABA, PABA and their derivatives can be based on the shikimate pathway, starting from a simple carbon source such as glucose and catalyzed by various enzymes. PABA and OABA are derived from chorismate (CHO) in the shikimate pathway, while the synthesis of MABA occurs before the CHO [8,21,22].

The synthesis of ABA and its derivatives via the shikimate pathway begins with glucose undergoing glycolysis to form phosphoenolpyruvic acid (PEP). Through the pentose phosphate pathway (PPP), it is further converted to erythrose-4-phosphate (E4P) (Figure 2). PEP and E4P are then used by the 3-deoxy-D-arabino-heptulosonate-7-phosphate (DAHP) synthase enzymes encoded by *aroF*, *aroG*, or *aroH* to synthesize DAHP [23]. DAHP is converted into the intermediate shikimate through the conversion of 3-dehydroquinate (DHQ) and 3-dehydroshikimate (DHS). Subsequently, under the hydrolysis of ATP and the introduction of a second PEP, shikimate passes through two intermediates, shikimate-3-phosphate (S3P) and 5-enolpyruvyl-shikimate-3-phosphate (EPSP), and is converted into a common precursor CHO of aromatic amino acids [24,25].

After the formation of CHO, the separated branches differentiate into biosynthesis of phenylalanine, tyrosine, and tryptophan. The enzyme anthranilate synthase (TrpEG), encoded by the *trpEG* gene, catalyzes the conversion of CHO into OABA and pyruvate, utilizing glutamine (Gln) as the amino donor. OABA serves as the first intermediate in the tryptophan biosynthesis pathway [25]. In addition to its role as anthranilate synthase, TrpD can also function as anthranilate phosphoribosyltransferase. It catalyzes the conversion of anthranilate to N-(5'-phosphoribosyl)anthranilate, which is an intermediate in the downstream metabolism. Subsequently, through several steps, this intermediate can be further converted into tryptophan [26]. Tryptophan can also serve as a precursor for many chemical substances and continue to be converted into downstream products such as indole and auxin through the tryptophan metabolism pathway [17]. In addition, OABA can also be catalyzed by anthranilic acid methyltransferase 1 (AAMT1) to produce the grape flavor compound methyl anthranilate [24,27]. Alternatively, OABA can be converted into catechol by specific enzymes (terminal oxygenase component (AntAB), and reductase component of anthranilate 1,2-dioxygenase (AntC)) through odihydroxylation, spontaneous deamination, and decarboxylation [28].

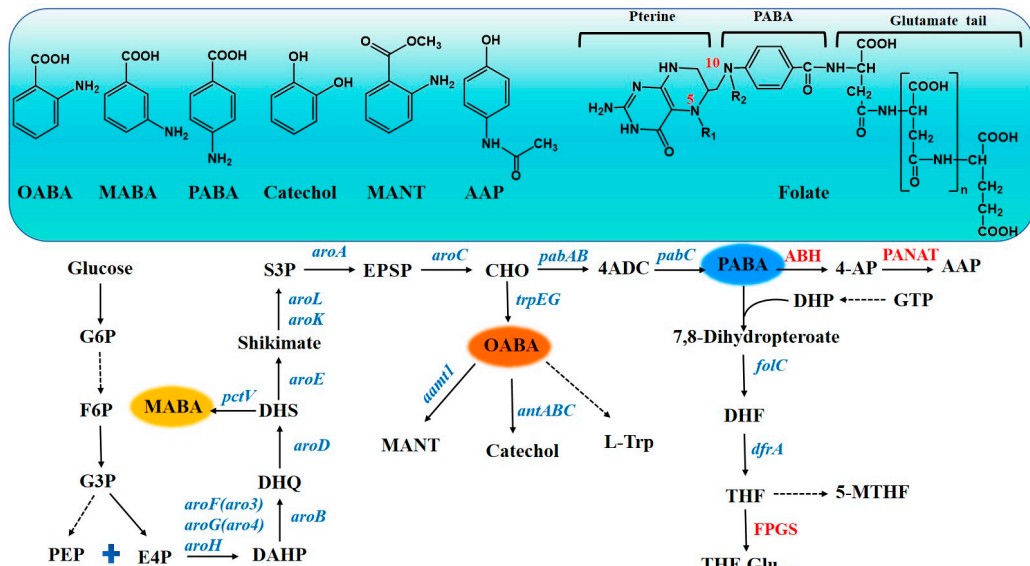

**Figure 2.** Biosynthetic pathways of aminobenzoic acid and its derivatives. G6P, glucose-6-phosphate; F6P, fructose-1,6-bisphosphate; G3P, glycerinaldehyde-3phosphate; PEP, Phosphoenolpyruvate; E4P, D-erythrose 4-phosphate; DAHP, 3-deoxy-D-arabino-heptulosonate-7-phosphate; DHQ, 3dehydro-quinate; DHS, 3-dehydroshikimate; MABA, meta-aminobenzoic acid; S3P, shikimate-3-phosphate; EPSP, 5-enolpyruvyl-shikimate-3-phosphate; CHO, chorismate; OABA, ortho-aminobenzoic acid; MANT, methyl anthranilate; L-Trp, L-tryptophan; 4ADC, 4-amino-4-deoxychorismic acid; PABA, para-aminobenzoic acid; DHF, dihydrofolate; THF, tetrahydrofolate; THF-Glu(n)—tetrahydrofolate polyglutamate; 5-MTHF, 5-methyltetrahydrofolate; 4-AP, 4-aminophenol; AAP, Acetaminophen/N-acetyl *p*-aminophenol; *aroF* (aro3), *aroG* (aro4), *aroH*, encoding DAHP synthase isoenzyme; *aroB*, encoding DHQ synthase; *aroD*, encoding DHQ dehydratase; ABH, 4-aminobenzoate hydroxylase; PANAT, arylamine N-acetyltransferase; FPGS, folylpolyglutamate synthase; *pctV*, encoding 3-aminobenzoate synthase; *aroE*, encoding shikimate dehydrogenase; *aroK*, encoding shikimate kinase; *aroL*, encoding shikimate kinase 2; *aroA*, encoding EPSP synthase; *aroC*, encoding chorismate synthase; *pabAB*, encoding 4-amino-4-deoxychorismic acid synthase; *pabC*, encoding aminodeoxychorismate; *trpEG*, encoding anthranilate synthase; aamt1, ANT methyltransferase1; *antABC*, encoding ANT 1,2-dioxygenase; *folC*, encoding folyl-polyglutamate synthetase; *dfrA*, encoding dihydrofolate reductase.

## *2.2. MABA*

The precursor substance of MABA acid is DHS, which is converted to MABA by MABA synthase (PctV) before being converted to shikimate [21]. There is currently limited research on the downstream metabolism of MABA; therefore, there is a lack of in-depth studies on its metabolic pathways (Figure 2).

## *2.3. PABA and Its Derivatives*

In the pathway of PABA synthesis, it begins with the catalysis of aminodeoxychorismate synthase subunit II (PabA), which releases the ammonia from the δ-amino group of glutamine. Then, aminodeoxychorismate synthase subunit I (PabB) catalyzes the replacement of the hydroxyl group on CHO with the released ammonia, resulting in the formation of 4-amino-4-deoxychorismic acid (4ADC) (Figure 2). Finally, 4ADC lyase (PabC) catalyzes the cleavage of 4ADC, leading to the release of a pyruvate molecule and the aromatization of the 4ADC ring, ultimately generating PABA [23,29]. In some microorganisms, *pabA* and *pabB* fuse to construct a *pabAB* gene, encoding a heterodimeric enzyme (4ADC synthesis) that can directly catalyze the synthesis of 4ADC, and the PabAB protein is more stable than the isolated PabAB protein, which is more conducive to the production of PABA [30].

PABA, as a precursor, can be converted into downstream products such as paracetamol (N-acetyl *p*-aminophenol (AAP)), folate, and its metabolites. The enzyme 4-aminobenzoate hydroxylase (ABH), along with the cofactor NADH regeneration system involving glu-

cose dehydrogenase (GDH), catalyzes the conversion of PABA to 4-aminophenol (4-AP). Arylamine N-acetyltransferase (PANAT) undergoes aminoacetylation of 4-AP to generate AAP [31].

PABA also has a very important derivative, folic acid. In living organisms, folate is synthesized from the 2-amino-4-hydroxy-pteridine, PABA, and glutamate [32] (Figure 2). The reaction of GTP to dihydroneopterin is catalyzed by GTP cyclohydrolase. Then, phosphatase removes a phosphate residue. After that, dihydroneopterin aldolase acts on the product to give glycolaldehyde and 6-hydroxymethyl-7,8-dihydropterin, which is converted to 6-hydroxymethyl-7,8-dihydropterin pyrophosphate (DHPPP) by 6-hydroxymethyldihydropteridine pyrophosphokinase. Dihydropteroate synthase pairs DHPPP to PABA. The C-N bond of DHPPP connects with pABA to form dihydropterin (DHP). DHP is then glutamylated with glutamate, and reduced by dihydrofolate reductase to yield the biologically active coenzyme tetrahydrofolate (THF). Finally, polyglutamate folate synthetase adds multiple glutamate moieties to THF to yield THF-polyglutamate [4,33]. Subsequent steps can be taken to synthesize other products of the folate family, such as 5-methyltetrahydrofolate [34].

## 3. Biosynthesis of OABA and Its Derivatives

### 3.1. OABA

OABA is a white to pale yellow solid compound with a slightly sweet taste. It has a melting point of 144–146 °C and can crystallize from hot water. Solutions in alcohols or ethers exhibit a violet crystalline-like fluorescence. Its fluorescent properties can be utilized for biological analysis purposes, such as monitoring protein glycosylation [3]. OABA has various functions, including anti-inflammatory, analgesic [35], antibacterial [36], and mood-improving effects [37]. Therefore, it is commonly used in the treatment of psychiatric disorders, inflammation, and other related diseases [38,39]. In addition, OABA can also be used as a compound precursor in the chemical and food industries, such as food additives, dyes, perfumes, crop protection compounds, plastics, and so on [40].

OABA is a major metabolic product in organisms and can be synthesized into the amino acid L-tryptophan, which is essential for protein synthesis. It is produced by diverting some of the cellular flux from its pathway of synthesizing phenylalanine and tyrosine from CHO [41]. The earliest attempt to synthesize OABA was in *E. coli*. Balderas Hernandez et al. [42] characterized the *trpD* gene mutant W3110 *trpD*9923 and modified it using metabolic engineering strategies. Glucose was used as a carbon source and fed batch fermentation was used to produce a yield of 14 g/L within 34 h (Table 1). In addition to *E. coli*, researchers have also studied the synthesis of OABA in other strains such as *Pseudomonas putida*, *Saccharomyces cerevisiae*, and *Corynebacterium glutamicum*. A modified *C. glutamicum* [25] strain using a mixture of glucose and xylose as a carbon source achieved a titer of 5.9 g/L in CGXII medium. However, the yields of the modified *Pseudomonas putida* [8] and *Saccharomyces cerevisiae* [27] were relatively lower, at 567.9 mg/L and 1.54 g/L, respectively. Nevertheless, these studies have explored new avenues for the sustainable production of OABA.

**Table 1.** Biosynthesis of aminobenzoic acid and its derivatives.

| Product | Host | Carbon Source | Titer (g/L) | Time | Fermentation Mode | Engineered Strategy | Advantages | Disadvantages | Reference |
|---------|------|---------------|-------------|------|-------------------|--------------------|-----------|---------------|-----------|
| OABA | *C. glutamicum* | Glucose and xylose | 5.9 | 32 h | Pulsed-fed-batch | 3-Deoxyarabinoheptulosonate-7 phase syntax and TrpEG were imported, and genes nagD and qsuD were removed | Engineering *C. glutamicum* with strong tolerance | Expensive and complex culture media | [25] |
| | *E. coli* | Glucose | 4.0 | 60 h | Fed-batch | Destruction of pheA, tyrA, pabA, ubiC, and entC, and trpR genes, overexpression of aroE and tktA genes | Higher yield | Complex strain transformation steps | [40] |
| | *S. cerevisiae* | D-glucose | 0.57 | 72 h | Batch | Disruption of TRP4, overexpression of GLN1, regulation of Aro4 and Trp2, and overexpression of TRP3 | Realize the production of OABA by *S. cerevisiae* | Low carbon flux | [27] |
| MANT | *S. cerevisiae* | D-glucose | 0.41 | 72 h | Batch | Expression of anthranilic acid methyltransferase 1 from *Medicago truncatula* | Realize the production of MANT by *S. cerevisiae* | Low carbon flux | [27] |
| | *C. glutamicum* | Glucose | 5.74 | 110 h | Fed-batch | Expression of pSH36HTc and pEKGH in *C. glutamicum* | The reaction is simpler and the cofactors are less. | High residual levels of precursor metabolites | [24] |
| | *E. coli* | Glucose | 4.47 | 72 h | Fed-batch | Expression of pBBR1G$^{fbr}$A$^{fbr}$E$^{fbr}$ and pTacT in *E. coli* | The reaction is simpler and the cofactors are less. | Low enzyme catalytic activity | [24] |
| Catechol | *E. coli* | Glucose | 4.47 | 76 h | Fed-batch | Expression of 1,2-dioxygenase, DAHP synthase and transketolase. in *E. coli* | Low cultivation cost | Lower glucose consumption capacity of the strain | [43] |
| MABA | *E. coli* | Glucose | 0.048 | 6 d | Batch | Coupling of 3AB synthase PctV with engineered shikimate pathway | The modular nature of co culture engineering allows for rapid identification of specific enzymes or optimal strains | Low activity of MABA synthase PctV | [21] |
| PABA | *E. coli* | Glycerol | 0.84 | 48 h | Batch | Strengthening the shikimate pathway and overexpressing PABA synthase | Few by-products | The enzyme promiscuity is not beneficial to the production | [44] |
| | *S. cerevisiae* | Glycerol-ethanol | 0.22 | 78 h | Fed-batch | Overexpression of ABZ1 and ABZ2 genes in wine yeast AWRI1631 and QA23 | Low process cost | High glycerol will have a negative impact on PABA titer | [22] |
| | *C. glutamicum* | Glucose | 43 | 48 h | Test-tube scale culture | Introducing pabAB from *C. callunae* and pabC from *X. bovienii* into strains overexpressing the shikimate pathway | High PABA production titer | *folP* deficiency can lead to malnutrition in bacterial strains | [29] |
| Folate | *Bacillus subtilis* | Glucose | 0.003 | 15 h | Fed-batch | Replacing yitJ with metF, knocking out purU, overexpressing dfrA, folC, pab, folE, and yciA, inhibiting thyA, pheA, trpE, and panB genes | Production cost reduction | Low yield | [45] |
| | *S. cerevisiae* | Glucose | 0.13 | 20 h | Batch | SIDA (Stable Isotope Dilution Analysis) determination and Molecular Network (MN) Analysis | Increasing of yield | C$_2$-metabolic mechanism unclear | [46] |
| Acetaminophen | *E. coli* | Glycerol | 0.12 | 48 h | Batch | Expression of *p*-AP N-acetyltransferase | Implementation of producing AAP with a simple carbon source | High by-products | [44] |
| | *E. coli* | Glucose | 0.94 | 24 h | Fed-batch | Heterologous expression of enzymes from five different microbial sources, modification of ABH and PANAT enzymes | The increase in AAP yield, no need for auxiliary factors | Medium with complex composition | [31] |

The microbial production of OABA is actually limited by the fact that OABA has a toxic effect on microorganisms. Therefore, enhancing the tolerance of the production host would have a huge impact on improving the yield of OABA [47]. Some auxiliary methods can also be considered. For example, Li et al. [26] used a biosensor to assist in cell selection and in situ product removal to enhance the synthesis of OABA. Fernandez-Cabezon et al. [7] achieved certain success in increasing the yield of OABA by introducing components of *Pantoea stewartii*'s Esa quorum sensing (QS) system into modified soil bacterium *Pseudomonas putida*, which also avoids the use of expensive additives.

### 3.2. Methyl Anthranilate

Methyl anthranilate (MANT) is a natural metabolite that imparts grape aroma and flavor. It is widely used in foods such as candy, flavor enhancers, and beverages, as well as in medicines. It is also an important ingredient in perfumes and cosmetics [48]. MANT can also be used in other industrial applications, such as bird and geese repellents for crop protection, as an antioxidant or sunscreen agent, and as an intermediate for the synthesis of various chemicals, dyes, and pharmaceuticals. However, the low content of MANT in plants makes it impractical to meet industrial demands through extraction methods [24]. There are two pathways for synthesizing MANT from OABA. One is activation of CoA catalyzed by anthranilate-CoA ligase, followed by acyl transfer through alcohol acyltransferase, which uses CoA, ATP, and methanol as co substrates [48]. Another route is to use S-adenosyl-L-methionine (SAM)-dependent methyltransferase to catalyze the one-step conversion of OABA to MANT [49]. The second approach is simpler, and currently researchers are more inclined to study this approach.

Luo et al. [24] constructed an MANT synthesis pathway derived from plants in *E. coli* and *C. glutamicum*. They optimized the expression of anthranilic acid methyltransferase 1 (AAMT1), increased the supply of the precursor OABA, and used an in situ two-phase extraction fermentation method with tributyrin as the extractant to overcome the toxicity of MANT. As a result, *E. coli* and *C. glutamicum* produced 4.47 g/L and 5.74 g/L of MANT, respectively. Kuivanen et al. expressed AAMT1 in *S. cerevisiae* and disrupted the TRP4 gene encoding OABA ribotransferase, ultimately producing 414 mg/L of MANT in YPD medium [27].

### 3.3. Catechol

Catechol, also known as 1,2-dihydroxybenzene, is an aromatic compound that is soluble in water, ethanol, benzene, chloroform, and pyridine. As a fine chemical raw material, it is widely used in the production of pharmaceuticals (isoproterenol), pesticides (propoxur and carbofuran), spices, dyes, and rubber [28]. There are many pathways for the biosynthesis of catechol, such as the DHS pathway [50] or the 4-hydroxybenzoic acid (4-HBA) pathway derived from CHO [51], as well as the OABA pathway. Perhaps the OABA pathway is not as simple as the first two pathways, and currently only Balderas-Hernández et al. [43] have studied the biological production of catechol through this pathway. They constructed a recombinant *E. coli* system capable of producing a large amount of OABA. Subsequently, the anthranillate 1,2-dioxygenase gene in *P. aeruginosa* PAO1 was expressed in the recombinant strain, reconstructing a new pathway for catechol production. A batch culture using glucose as a carbon source produced 4.47 g/L of catechol.

## 4. Biosynthesis of MABA

MABA is a commonly used derivative of aniline monomers and has made significant contributions in pharmacology and biology. As an indispensable pharmaceutical intermediate, it is widely used in the synthesis of painkillers, antihypertensive drugs, vasodilators, etc. [5]. In addition, it has great applications in the synthesis of new organic luminescent materials, electrochemical materials, azo dyes, plant growth regulators, and so on [52,53]. Although the application of MABA is so extensive and its synthesis in the shikimate pathway has been explored, there is currently little research on the biosynthesis of MABA [54,55].

Currently, only Zhang and Stephanopoulos [21] have established a MABA biosynthesis pathway in *E. coli* by coupling the MABA synthase Pctv with the shikimate pathway. This pathway has been modularized into upstream and downstream strains, and an *E. coli–E. coli* co culture system has been constructed. With glucose substrate, the MABA production reached 48 mg/L. There are too few research examples on the biosynthesis of MABA, and there have been no reports on related synthesis in recent years, which has led to a lack of deeper achievements in MABA biosynthesis. Perhaps we can refer to the biosynthesis methods of OABA and PABA, and try to produce MABA by using *C. glutamicum*, *Saccharomyces cerevisiae*, or exploring better metabolic pathways and more effective enzymes.

## 5. Biosynthesis of PABA and Its Derivatives

### 5.1. PABA

PABA is a widely present aromatic substance in bacteria, fungi, plants, and some parasites, but it does not exist in humans and animals. It has a wide range of biological, industrial, and pharmaceutical applications [56,57]. Although it is not a vitamin, it is still referred to as a member of vitamin B. A lack of PABA can lead to diseases such as white hair, fatigue, depression, and irritability. Therefore, it is often used as a precursor and therapeutic agent in the pharmaceutical industry [58]. In addition, PABA is also used as a dye and feed additive, and can serve as a basic component of crosslinking agents for polyurethane resins. It has the potential to be a basic component of aromatic polymers [23].

PABA is generally synthesized in two steps from shikimate in various organisms, as described earlier. Based on the shikimate pathway and combined with genetic engineering, artificial biosynthesis of PABA has been achieved. Aversch et al. improved *Saccharomyces cerevisiae* using genetic engineering and metabolic models, overexpressing the ABZ1 and ABZ2 genes of *Saccharomyces cerevisiae* AWRI1631 and QA23. Using glycerol-ethanol as a composite carbon source, a PABA of 215 mg/L was obtained [22]. Koma et al. [23] introduced *aroF^{fbr}*, *pabA*, *pabB*, and *pabC* genes controlled by the T7lac promoter into *E. coli* chromosomes for metabolic engineering modification, resulting in excessive production of PABA from glucose, reaching 4.8 g/L. Shen et al. [44] also utilized *E. coli* to produce 836.43 mg/L of PABA during the construction of AAP, with a relatively low yield. Like OABA, in the process of microbial production of PABA, we must consider the toxicity of PABA to microorganisms because this will greatly affect the yield of PABA [59,60]. Kubota et al. [29] selected the most tolerant *C. glutamicum* after evaluating the PABA toxicity sensitivity of several microorganisms. The *pabAB* gene of *C. calunae* and the *pabC* gene of *X. bovienii* were overexpressed in *C. glutamicum*. Under fermentation control conditions, the strain produced 43 g/L PABA within 48 h. It is the highest yield so far.

### 5.2. Folate

Folate, also known as folic acid, is a group of over 150 heterocyclic compounds with similar structures [61]. They belong to the vitamin B9 category as a whole. The most stable form of these vitamins is fully oxidized pteroylglutamic acid (PteGlu). Due to the different oxidation states within the pterin ring, different 1C-units connected at positions $N^5$ and/or $N^{10}$ (typically 1C-units in the form of methyl ($CH_3$-), formyl (HCO-), or methenyl (-$CH^+$-) groups), and changes in the length of the glutamyl tail, a large number of different forms of folates are produced [46] (Figure 2). Many physiological processes, such as the biosynthesis of purine and thymidine monophosphate, the regeneration of methionine and the translation of mitochondrial protein, require folate for 1C-unit transfer [61]. All these reactions are essential for maintaining the health function of organisms. However, animals lack the biosynthetic pathway of folate, so it is necessary to provide an appropriate amount of these compounds through food intake [62]. If there is a lack of intake of folate, it can increase the risk of anemia, and cardiovascular and neurological diseases, especially leading to fetal neural tube defects [63].

Many microorganisms themselves can produce folate, such as lactic acid bacteria such as *Lactococcus lactis*, *Streptococcus thermophilus*, *bifidobacteria*, yeast, etc., but their production

can only reach the level of micrograms based solely on their own metabolism [64,65]. For example, the Kefir yeast strains isolated by Patring et al. had an average folate production of only 43 μg/L [66]. This level of production cannot meet the needs of industrial production, and research has found that metabolic engineering and synthetic biology can increase the production of folate. Serrano-Amatriain et al. [67] overexpressed the FOL gene in the industrial fungus *Ashbaya gossypii* and increased folate production to 6.595 mg/L using combinatorial coordination engineering. Yang et al. [45] transferred metabolic flux to the biosynthesis of 5-methyltetrahydrofolate (5-MTHF) in *Bacillus subtilis* by replacing the natural *yitJ* gene with *E. coli metF*, knocking out *purU*, and overexpressing *dfrA*. The co-overexpression of *folC*, *pabB*, *folE*, and *yciA* enhanced the supply of 5-MTHF precursors and inhibited the *pheA* gene, successfully increasing the titer of 5-MTHF to 1.58 mg/L. Schillert et al. [46] applied the molecular network to the folate metabolism of yeast. After 20 h of culture, the folate content reached the highest, which was 131.2 mg/L.

From the above research, it can be seen that even with the use of advanced technologies such as cell engineering and synthetic biology, the yield of folate is still relatively low compared to other derivatives of ABA, and there is still a lot of room for improvement. Perhaps due to the instability of folate itself and its sensitivity to spontaneous and photo oxidative degradation, the final yield of folate obtained is not high [68]. Therefore, reasonable avoidance of consumption of these may to some extent increase folate production.

### 5.3. Acetaminophen

Acetaminophen (AAP), whose chemical compound name is N-acetyl *p*-aminophenol, also known as paracetamol, is a widely used non-NSAID analgesic and one of the most widely used and safe over-the-counter (OTC) drugs for treating pain and fever. It is the preferred drug for patients who cannot be treated with nonsteroidal anti-inflammatory drugs (NSAID) [69,70]. It once played a huge role in the treatment of fever during the period of COVID-19 [32].

Initially, AAP was produced in large quantities as a pharmaceutical raw material through chemical synthesis [71]. With the development of research, it was isolated as a natural product, but its synthetic gene cluster and pathway were not yet clear at that time [72]. It was not until recent years that the biosynthetic pathway of AAP was discovered, and its synthesis methods have taken on new directions. Shen et al. [44] identified and characterized a PABA monooxygenase and a p-aminophenol (PAP) N-acetyltransferase, which can convert PABA to PAP and PAP to AAP, respectively. By strengthening the shikimate pathway in *E. coli* and using glycerol as the carbon source, 120.03 mg/L AAP was ultimately obtained. Hou et al. [31] used *E. coli* as the host and carried out protein engineering modifications on a FAD dependent monooxygenase, 4-aminobenzoate hydroxylase (ABH) and arylamine N-acetyltransferase (PANAT), using glucose or glycerol as the carbon source to produce 0.94 g/L AAP in a 5 L feed fermentation tank. Using PABA as the substrate, 4.2 g/L AAP was obtained. From the perspective of synthetic pathways and enzyme functions, ABH and PANAT are similar to PABA monooxygenase and PAP N-acetyltransferase, respectively. Therefore, the enzyme used by Hou and Shen et al. should be the same.

## 6. Conclusions

ABA and its derivatives, as aromatic compounds produced through the shikimate pathway, play a prominent role that is closely related to our lives and is indispensable. However, synthesizing these compounds from non-renewable petroleum extracts is not a sustainable solution. Moreover, this production method poses irreversible damage to the environment, including the pollution of soil, water bodies, and air, which is detrimental to all forms of life on Earth. Utilizing simple carbohydrates as carbon sources and employing microbial activities to synthesize the required ABA and its derivatives is an excellent alternative. This approach can not only address the issue of unsustainable petroleum resources but also significantly mitigate environmental pollution.

Biological synthesis is currently a relatively mature technology, and the synthetic pathways of ABA and its derivatives in plants and microorganisms have been elucidated. However, based on the current literature description, further research is needed for the biosynthesis of ABA and its derivatives. For example, there is little research on the biosynthesis of MABA, and the low yield of folate biosynthesis urgently needs to be addressed. The biosynthesis methods of ABA and its derivatives summarized earlier have achieved microbial autonomous synthesis, but they are still at the laboratory level, and there is still a certain distance from industrial large-scale production. The obstacle in this step is that the yield of various microorganisms cannot meet the requirements of industrialization. At present, continuing to optimize metabolic engineering strategies, improve microbial yield, and lay a solid foundation for industrial production is of utmost importance. Therefore, the future development direction may be explored from the following aspects: first, one must search for or cultivate more suitable strains with stronger resistance to the toxicity of ABA and its derivatives, in order to adapt to their toxic nature. Secondly, one must recognize that the biosynthesis of each compound may not be limited to a single pathway, and there may exist simpler branches. If these branches can be identified, it would increase synthesis efficiency. Thirdly, attention should be paid to whether the use of genetic engineering to enhance or inhibit the expression of certain enzyme genes, or to cut off certain gene segments, will have an inhibitory effect on certain feedback effects during the synthesis process, because some products in the biosynthetic pathway will in turn inhibit or promote the synthesis effect. Fourthly, waste materials such as lignin can be utilized as much as possible to create new value in biosynthesis, while also avoiding waste and environmental pollution caused by the accumulation or disposal of these wastes.

**Author Contributions:** Writing—original draft preparation, S.X., R.Z. and B.W.; writing—review and editing, S.X., S.Z., J.Z. and J.C.; visualization, J.C.; supervision, J.C.; project administration, S.Z., J.C. and J.Z.; funding acquisition, J.C. and J.Z. All authors have read and agreed to the published version of the manuscript.

**Funding:** This work was supported by the earmarked fund for CAR-43, the National Modern Agricultural Industrial Technology System, Sichuan Innovation Team Construction Project (SCSZTD-2022-08-07), the Open Funding Project of Meat Processing Key Laboratory of Sichuan Province (23-R-06), and partially supported by Solid-state Brewing Technology Innovation Center of Sichuan.

**Conflicts of Interest:** Authors Suyi Zhang and Jie Cheng were employed by Luzhou Laojiao Co., Ltd. The remaining authors declare that the research was conducted in the absence of any commercial or financial relationships that could be construed as a potential conflict of interest.

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
