# Peer review of "A New Direction for the Green, Environmentally Friendly and Sustainable Bioproduction of Aminobenzoic Acid and Its Derivatives"

_sustainability, doi:10.3390/su16073052_

Round 1

Reviewer 1 Report

Comments and Suggestions for Authors

This manuscript provided a good review of sustainable synthesis of aminobenzoic acid and derivatives. The manuscript should address the following issues to be considered for publication.

1. The English language needs to be extensively improved to be more clear to avoid confusion to the readers. 

2. Please go through the manuscript to correct any typos, for example line 85 "PABA)".

3. Please provide a higher resolution Figure 1.

4. In line 98 and 99, the authors stated TrpD catalyzed CHO conversion to OABA, but in Figure 2 it's trpEG. Also, the authors used abbreviations in Figure 2 and caption, but would be better if abbreviation were added to the text after full name to be more readable.

5. References should be consistent in format.

Comments on the Quality of English Language

English language quality needs to be improved. 

Author Response

Reviewer #1

This manuscript provided a good review of sustainable synthesis of aminobenzoic acid and derivatives. The manuscript should address the following issues to be considered for publication.

Comment 1:The English language needs to be extensively improved to be more clear to avoid confusion to the readers.

Response: Thank you for your comments. Now the language was improved in the revised paper.

Comment 2:Please go through the manuscript to correct any typos, for example line 85 "PABA)".

Response: Thank you for your comments. In lines 97, 284, 288, we have corrected the spelling errors in the article

Comment 3:Please provide a higher resolution Figure 1.

Response: Thank you for your comments. In page 2, we have replaced it with a higher resolution Figure 1.

Comment 4:In line 98 and 99, the authors stated TrpD catalyzed CHO conversion to OABA, but in Figure 2 it's trpEG. Also, the authors used abbreviations in Figure 2 and caption, but would be better if abbreviation were added to the text after full name to be more readable.

Response: Thank you for your comments.

In line 110, 111, “TrpD” and “trpD” have been replaced by “TrpEG” and “trpEG”. In addition, the use of the full names of various compounds in the figure will be very messy. The full names of various compounds have been placed in the form of annotations after the title of Figure 2 for easy one-to-one reference.

Comment 5:References should be consistent in format. 

Response: Thank you for your comments. The format of the references has been revised in the revised manuscript.

Reviewer 2 Report

Comments and Suggestions for Authors

The Manuscript by Zhang et al. is a Review concerning the biological production (mainly microbiological) of aminobenzoic acids; it includes 70 reference sources from 1995 till 2023, mainly literature by recent 5 years. In general, I think that this subject may be interesting to the Readers of Sustainability journal. At the same time the style of writing is too general (statement of facts, low level of generalization). The main drawback is the absence of chemical formulas and structures; without structures and reaction schemes it is very difficult to percept the chemistry performed (add Schemes of chemical transformations). Moreover, there are several phrases in the text (like ll.31-32) which are well known and don't suit the style of scientific publication. So, I recommend Major Revision concerning mainly the presentation of the Material for a wider audience.

Additional Comments:

1)      Unsuccessful phrase (l.12-13: “Its main source (?) is the chemical synthesis of benzene (?) derived (?)…” etc.).

2)      L.20: should be “para-aminobenzoic acid” etc.

3)      L. 29: “… ABA is divided into three isomers” is incorrect chemically. Compound cannot be divided into isomers! For one substance several isomer may exist. Furthermore, anthranilate is not isomer of ABA.

4)      L.76: after reading I didn’t percept the “progress on the synthesis of ABA”. Apparently, I cannot understand the chemistry improving.

5)      Ll. 154-155: “… has three main structures…” This phrase is incorrect chemically. There are no partial structures in the compound; there are substituents or groups (pteridinyl, glutamate etc.). Without insertion of chemical structure it is impossible to understand the content!

6)      L.280: “ ….they are composed of three parts…”. It’s incorrect (see above)!

7)      L. 283: N5 and N10. Which ones?

8)      Misprints: ll.85, 305;

9)      L. 316: “scientific name”?! The name of chemical compound.

10)   In reference [24] the source was omitted.

Comments on the Quality of English Language

Minor editing of English language required.

Author Response

Reviewer #2

Comment 1:The Manuscript by Zhang et al. is a Review concerning the biological production (mainly microbiological) of aminobenzoic acids; it includes 70 reference sources from 1995 till 2023, mainly literature by recent 5 years. In general, I think that this subject may be interesting to the Readers of Sustainability journal. At the same time the style of writing is too general (statement of facts, low level of generalization). The main drawback is the absence of chemical formulas and structures; without structures and reaction schemes it is very difficult to percept the chemistry performed (add Schemes of chemical transformations). Moreover, there are several phrases in the text (like ll.31-32) which are well known and don't suit the style of scientific publication. So, I recommend Major Revision concerning mainly the presentation of the Material for a wider audience.

Response: Thank you for your comments. We have added the chemical structural formula to the path diagram, and made modifications and additions to the statement in the article

Comment 2:Unsuccessful phrase (l.12-13: “Its main source (?) is the chemical synthesis of benzene (?) derived (?)…” etc.).

Response: Thank you for your comments. 

In line 12-14, We have replaced “Its main source is the chemical synthesis of benzene derived from petroleum” with the fluent phrase “Their production relies on chemical synthesis using petroleum derived substances such as benzene as precursors”.

Comment 3:L.20: should be “para-aminobenzoic acid” etc.

Response: Thank you for your comments. We have corrected the incorrect description here.

Comment 4:L. 29: “… ABA is divided into three isomers” is incorrect chemically. Compound cannot be divided into isomers! For one substance several isomer may exist. Furthermore, anthranilate is not isomer of ABA.

Response: Thank you for your comments.We have corrected the incorrect description here.

Comment 5:L.76: after reading I didn’t percept the “progress on the synthesis of ABA”. Apparently, I cannot understand the chemistry improving.

Response: Thank you for your comments. The purpose of this paper is to summarize and describe an ABA biosynthesis method in parallel with chemical synthesis, rather than to improve some of the original chemical synthesis methods. Because of the environmentally friendly characteristics of biosynthesis, it can replace chemical synthesis to a certain extent. Perhaps the author did not express their intention clearly in the article, so additional descriptions have been made in the introduction section.

Comment 6:L. 154-155: “… has three main structures…” This phrase is incorrect chemically. There are no partial structures in the compound; there are substituents or groups (pteridinyl, glutamate etc.). Without insertion of chemical structure it is impossible to understand the content!

Response: Thank you for your comments. In line 166-167, We have modified the error description. And we have added the chemical structural formula to the path diagram

Comment 7:L.280: “ ….they are composed of three parts…”. It’s incorrect (see above)!

Response: Thank you for your comments. In line 296-297, We have modified the error description.

Comment 8:L. 283: N5 and N10. Which ones?

Response: Thank you for your comments. In line 299-301, we have replaced it with a more understandable sentence.

Comment 9:Misprints: ll.85, 321;

Response: Thank you for your comments. In line 97 and 310, we have corrected the mistakes.

Comment 10:L. 332: “scientific name”?! The name of chemical compound.

Response: Thank you for your comments. In line 321, we have made modifications to the inappropriate descriptions.

Comment 11:In reference [24] the source was omitted.

Response: Thank you for your comments. In line 458-460, we have added the source of the eference.

Reviewer 3 Report

Comments and Suggestions for Authors

The submitted manuscript aims to address the obtaining of aminobenzoic acid and its derivatives. The title suggests that the manuscript's theme also covers chemical synthesis processes. However, a reasonably brief description is made of its biosynthesis processes, isomers, and derivatives using microorganisms. There are too many references in the literature regarding chemical synthesis. Below are some of them:

https://pubs.rsc.org/en/content/articlelanding/2015/ra/c5ra04950d

https://www.mdpi.com/2227-9059/11/10/2686

It needs to be clarified why these types of investigations and reports were ignored. There must be evidence of coherence between the manuscript's content and its title.

Below, I present other comments related to the content of the manuscript:

1. Reaction schemes must be distinguished from figures, such as Figure 2, which must be constructed with good resolution and correct chemical structures. It is generally evident that schemes are avoided and that chemical structures are not used in the document, making it easier to read.

2. It is mentioned that MABA does not need more in-depth studies on its metabolic pathways, but it does not relate or describe the studies carried out to date in greater depth.

3. Chemical structures must be used to illustrate the biosynthetic processes described.

4. It refers to many chemical compounds without indicating their stereoisomerism, for example, tryptophan and its chirality; which enantiomer do you refer to?

5. Generally, the text gives relevance to biotechnological processes but does not highlight the innovation or improvement of each process compared with the typical synthesis process from hydrocarbon derivatives. That should be the main contribution of the manuscript.

Given these observations, I consider that the current version of the manuscript should not be published.

Author Response

Comment 1:The submitted manuscript aims to address the obtaining of aminobenzoic acid and its derivatives. The title suggests that the manuscript's theme also covers chemical synthesis processes. However, a reasonably brief description is made of its biosynthesis processes, isomers, and derivatives using microorganisms. There are too many references in the literature regarding chemical synthesis. Below are some of them:

https://pubs.rsc.org/en/content/articlelanding/2015/ra/c5ra04950d

https://www.mdpi.com/2227-9059/11/10/2686

It needs to be clarified why these types of investigations and reports were ignored. There must be evidence of coherence between the manuscript's content and its title.

Response: Thank you for your comments. We have cited the above two references in the article

Comment 2:Reaction schemes must be distinguished from figures, such as Figure 2, which must be constructed with good resolution and correct chemical structures. It is generally evident that schemes are avoided and that chemical structures are not used in the document, making it easier to read.

Response: Thank you for your comments. Figure 2 is not a reaction plan, but a metabolic pathway based on the shikimate pathway, such as shikimate, chorismate, and even aminobenzoic acid, which can be self produced in organisms. The significance of the path diagram in Figure 2 is to express the production process of three aminobenzoic acids and their derivatives in the shikimic acid pathway. We can control the synthesis of downstream substances in this pathway by controlling the enzymes on this pathway, such as expressing or deleting certain enzyme genes. And we have added the chemical structural formula to the path diagram

Comment 3:It is mentioned that MABA does not need more in-depth studies on its metabolic pathways, but it does not relate or describe the studies carried out to date in greater depth.

Response: Thank you for your comments. The description in this paper does not mean that the metabolism of MABA does not require more in-depth research. It is precisely because there are too few related biosynthesis studies at present that the biosynthesis of MABA has not been further developed for so many years.

In line 263-268, we have provided more descriptions of the current status of MABA biosynthesis and put forward some views on future research directions

Comment 4:Chemical structures must be used to illustrate the biosynthetic processes described.

Response: Thank you for your comments. We have added the chemical structural formula to the path diagram

Comment 5:It refers to many chemical compounds without indicating their stereoisomerism, for example, tryptophan and its chirality; which enantiomer do you refer to?

Response: Thank you for your comments. In line 131 and 191, The tryptophan in the text refers to L-tryptophan, and we have corrected the relevant description.

Comment 6:Generally, the text gives relevance to biotechnological processes but does not highlight the innovation or improvement of each process compared with the typical synthesis process from hydrocarbon derivatives. That should be the main contribution of the manuscript.

Response: Thank you for your comments. In line 78-83, the comparison of chemical synthesis and biosynthesis has been added.

Reviewer 4 Report

Comments and Suggestions for Authors

The manuscript submitted by Shujian Xiao et al. offers valuable insights into the biosynthesis of ABA and its derivatives, addressing the pressing need for sustainable alternatives to traditional chemical synthesis methods. The comprehensive review of biosynthesis pathways, along with the analysis of examples and future directions, makes it a valuable resource for researchers in the field. Therefore, I recommend it can be published after the following questions being addressed.

1. The authors summarized the varying biosynthesis pathways. What are the advantages and disadvantages of each biosynthesis method mentioned? It will be better to list them in Table 1.

2. Have any of the biosynthesis methods discussed in the review been successfully scaled up for industrial production? What are the key challenges or barriers hindering the widespread adoption of biosynthesis methods in industrial settings? These points are recommended to be discussed in the revised manuscript.

3. In the analysis of biosynthesis methods, could the environmental benefits be quantified compared to traditional chemical synthesis approaches?

Author Response

Comment 1:The authors summarized the varying biosynthesis pathways. What are the advantages and disadvantages of each biosynthesis method mentioned? It will be better to list them in Table 1.

Response: Thank you for your comments. We have added the advantages and disadvantages of each biosynthesis in Table

Comment 2:Have any of the biosynthesis methods discussed in the review been successfully scaled up for industrial production? What are the key challenges or barriers hindering the widespread adoption of biosynthesis methods in industrial settings? These points are recommended to be discussed in the revised manuscript.

Response: Thank you for your comments. We have added the discussion on industrial large-scale production of biosynthesis to the conclusion.

Comment 3:In the analysis of biosynthesis methods, could the environmental benefits be quantified compared to traditional chemical synthesis approaches?

Response: Thank you for your comments. It is difficult to quantify environmental benefits. Chemical synthesis may be toxic to the environment, as well as harmful to human health, due to the presence of raw materials and by-products. Chemical synthesis often requires harsh conditions such as high temperature and high pressure, which can easily generate greenhouse gases. The conditions for biosynthesis reactions are mild, and raw materials and by-products often have no toxicity.

Round 2

Reviewer 2 Report

Comments and Suggestions for Authors

All questions are answered. The Manuscript is ready for publication.

Comments on the Quality of English Language

Only minor editing of English language required

Author Response

Thank you for your affirmation and comments. Now the language was improved in the revised paper. 

Reviewer 3 Report

Comments and Suggestions for Authors

The authors made some improvements to their manuscript. However, they were not significant improvements that could improve the technical-scientific quality of the manuscript. This document corresponds to a set of methods for obtaining aminobenzoic acid and derivatives from a biotechnological perspective and not at all with respect to chemical synthesis. I suggest changing the title of the manuscript by removing the term “synthesis”, which is defined by IUPAC as the construction of complex chemical compounds from simpler ones.

Author Response

Thank you for your comments. We have made revisions to the grammar and sentence structure of the entire text, and reviewed the biosynthesis of aminobenzoic acid and its derivatives. We have changed the word“synthesis” to “bioproduction”in title.

Round 3

Reviewer 3 Report

Comments and Suggestions for Authors

No have more comments.